# MODEL-BASED VALUE EXPLORATION IN ACTOR-CRITIC DEEP REINFORCEMENT LEARNING

## ABSTRACT

Off-policy method has demonstrated great potential on model-free deep reinforcement learning due to the sample-efficient advantage. However, it suffers extra instability due to some mismatched distributions from observations. Model-free on-policy counterparts usually have poor sample efficiency. Model-based algorithms, in contrast, are highly dependent on the goodness of expert demonstrations or learned dynamics. In this work, we propose a method which involves training the dynamics to accelerate and gradually stabilize learning without adding sample-complexity. The dynamics model prediction can provide effective target value exploration, which is essentially different from the methods on-policy exploration, by adding valid diversity of transitions. Despite the existence of model bias, the model-based prediction can avoid the overestimation and distribution mismatch errors in off-policy learning, as the learned dynamics model is asymptotically accurate. Besides, to generalize the solution to large-scale reinforcement learning problems, we use global gaussian and deterministic function approximation to model the transition probability and reward function, respectively. To minimize the negative impact of potential model bias brought by the estimated dynamics, we adopt one-step global prediction for the model-based part of target value. By analyses and proofs, we show how the model-based prediction provides value exploration and asymptotical performance to the overall network. It can also be concluded that the convergence of proposed algorithm only depends on the accuracy of learnt dynamics model.

## 1 INTRODUCTION

Model-free reinforcement learning (RL) algorithms have been applied to a wide range of tasks, ranging from simple games (Mnih et al., 2013; Oh et al., 2016) to robotic locomotion skills (Schulman et al., 2015). To tackle the large-scale continuous control problems, the function approximators implement some neural networks to represent the high-dimensional state and action spaces in deep reinforcement learning (DRL). However, model-free DRL is notoriously expensive in terms of its sample efficiency, which is deadly difficult to be employed in reality where samples are valuable to achieve. Among the recent model-free DRL algorithms, on-policy methods (Schulman et al., 2015; 2017; Fujimoto et al., 2018) typically require multiple samples to be collected for each rollout at every gradient step, which is quite extravagant in consuming samples because multiplied data requirement does not necessarily bring corresponding performance gain. In comparison, off-policy methods aim to reuse the past experience by storing the collected observations in a memory buffer, typically, combining Q-learning with neural networks (Mnih et al., 2015). Unfortunately, the combination of off-policy learning and high-dimensional, nonlinear function approximation are exposed to issues in terms of instability and divergence (Maei et al., 2009).

The causes for the emergent problems are very complicated, for example, some works (Fujimoto et al., 2018; 2019; Duan et al., 2021) blame them on the overestimation bias, which says the continually maximized value during the actor-critic optimization will accumulate the overestimation errors and break the training stability. Some others try to find extrapolation error induced by the mismatch between the distribution of sampled data from experience and true state-action visitation of the current policy (Fujimoto et al., 2019). There have been several ways to tackle the distribution mismatch. The authors in (Wu et al., 2019) address the distribution errors by extra value penalty or policy regularization, (Wang & Ross, 2019) changes the rule of experience replay to reduce the

distribution mismatch by sampling more aggressively from recent experience while ordering the updates to ensure that updates from old data do not overwrite updates from new data, and (Martin et al., 2021) relabels successful episodes as expert demonstrations for the agent to match. Despite their efforts, the overestimation bias and mismatched distribution from past experience can only be mitigated, and sometimes may induce new problems.

The paper has the following contributions. First, instead of using immediate rewards or assuming known reward function, we adopt neural networks to approximate the reward function as part of dynamics. Meanwhile, we train the parameters of modeled transition probability and reward function based on the replay buffer from off-policy observations. Second, the prediction from the learned dynamics will be used to foresee the target value according to a certain percentage. Since the dynamics-prediction is essentially different from the observations from environment, it can provide extra exploration which is not conditioned on the state-action visitation history. Besides, a well trained dynamics model is free of overestimation and distribution mismatch errors, and can provide more accurate target value and stabilize the asymptotic performance. Third, the related algorithm is proposed and the final results prove good efficiency and stability of the proposed algorithm. Fourth, the accuracy of learned model is tested by setting a maximum online time step, which is the beginning of off-line planning that is isolated from the environment.

## 2    RELATED WORK

Due to the various problems arising from the sample complexity of model-free algorithms, task-specific representations (Peters et al., 2010; Deisenroth et al., 2013) as well as the model-based algorithms (Deisenroth & Rasmussen, 2011; Levine et al., 2016; 2018; Kaiser et al., 2019) using planning, which optimize the policy under a learned or given dynamics model, are more preferable in real physical systems, such as robots and autonomous vehicles. However, task-specific representations have limited range of learnable tasks and greater requirement for domain knowledge. Model-based DRL algorithms are considered being more efficient (Deisenroth et al., 2013), because it constructs a dynamic probabilistic model via lots of data and avoids interaction with the environment by training the strategy based on the learned dynamics model (Hua et al., 2021), but it limits the policy to only be as good as the learned model (Gu et al., 2016).

For the model-free part, the agent needs to interact with the environment to collect enough knowledge for training, which poses the importance of the tradeoff between exploration and exploitation (Mnih et al., 2016). Soft actor-critic (SAC) (Haarnoja et al., 2018a;b) achieves good performance on a set of continuous control tasks by adopting stochastic function approximation and maximum entropy for policy exploration. Among these techniques, stochastic policies have the advantage of allowing on-policy exploration and off-policy experience replay over deterministic counterparts (Heess et al., 2015), and the maximum entropy exploration improves robustness and stability (Ziebart et al., 2008; Ziebart, 2010). Overall, the existing exploration strategies are limited to the policy, which raises the concern about whether and how the value exploration can play a positive role in model-free learning.

While some works combine both model-free and model-based DRL in the literature (Sutton, 1990; Lampe & Riedmiller, 2014), the following works are particularly relevant to our work in this paper. Specifically, (Gu et al., 2016; Nagabandi et al., 2018) add synthetic imagination rollouts to an additional replay buffer for model-guided exploration in some off-policy methods at the price of much higher storage and computation costs. Besides, model ensembles are adopted in (Chua et al., 2018; Kurutach et al., 2018; Janner et al., 2019) to reduce misguided policy or inaccurate planning caused by model bias. Moreover, value expansion of fixed multi-step prediction by dynamics model is adopted in (Feinberg et al., 2018; Buckman et al., 2018) to make proper value expansion and control imagination depth. However, making multi-step prediction from a global dynamics model may suffer cumulative model estimation errors and is usually replaced by iteratively refitted time-varying linear models (Levine & Abbeel, 2014). VIME (Houthooft et al., 2016) introduces maximization of information gain about the dynamics' certainty, which is overwhelmed by theoretical analyses and lack a bit intuitive judgement. In this paper, we adopt one-step prediction, which avoids the costs of storage and computation from multi-step synthetic rollouts, from a global dynamics model used for value exploration to achieve diversity, accuracy and generality.

## 3 PRELIMINARIES

We consider the extended Markov Decision Process (MDP) in continuous state and action spaces, denoted by the tuple $(\mathcal{S}, \mathcal{A}, P, r)$ where $\mathcal{S}$ is the state space, $\mathcal{A}$ is the action space, $\mathcal{S}$ is the space of next state, $P(s'|s, a)$ is the transition denoting the conditional probability of the next state $s' \in \mathcal{S}$ given the current state $s \in \mathcal{S}$ and action $a \in \mathcal{A}$, and $r \in \mathcal{S} \times \mathcal{A}$ is the reward function which is connected with $(s, a, s')$ from the environment. The purpose of reinforcement learning (RL) is to optimize the policy by maximizing the reward return, which is denoted by the expected discounted cumulative reward of a rollout. DRL employs the function approximation to parameterize the reward return so that the optimization can work under the setting of continuous control. In DRL, an action $a$ is sampled from the policy $\pi$ and "judged" according to a value estimate determined by the observation $s$, then the next state $s'$ and the future reward $r$ can be determined by the transition probability $P(\cdot|s, a)$ and the reward function $r(s, a)$, respectively, which are combined to define the dynamics in this work. In recent researches of DRL, the action-value (Q-value) function with respect to the state-action pair is usually chosen as the surrogate of the reward return, in the form of

$$Q_\pi(s, a) = \sum_t \mathbb{E}_{s_t \sim P^t, a_t \sim \pi} \left[ \gamma^t r(s_t, a_t) | s_0 = s, a_0 = a \right], \tag{1}$$

where $\gamma \in (0, 1)$ is the discount horizon factor for future rewards, $\pi$ is the policy for action section at every time step, and $P^t$ is the distribution of $s_t$, which is a joint distribution of transitions. If conditioned on initial state-action pair $(s_0, a_0)$, it is given by

$$P^t(s_t|s_0, a_0) = P(s_1|s_0, a_0) \prod_{i=1}^{t-1} \mathbb{E}_{s_i \sim \mathcal{S}, a_i \sim \pi} P(s_{i+1}|s_i, a_i). \tag{2}$$

Then we have

$$P^t(s_t) = \mathbb{E}_{s_{t-1} \sim \mathcal{S}, a_{t-1} \sim \pi} \left[ P(s_t|s_{t-1}, a_{t-1}) P^{t-1}(s_{t-1}) \right]. \tag{3}$$

**Lemma 1.** *Assume the expected KL-divergence between two transition distributions is bounded by*

$$\max_t \mathbb{E}_{a_t \sim \pi, s_t \sim P^t} D_{TV}(p(s_{t+1}|s_t, a_t)||P(s_{t+1}|s_t, a_t)) \leq \delta, \tag{4}$$

*then we have* $\mathbb{E}_{s_t \sim \mathcal{S}} |p^t(s_t) - P^t(s_t)| \leq 2t\delta$.

*Proof* See Appendix A (submitted in the supplementary material). □

The Q-value function in Eq. (1) is a mapping from the input observation-action pair $(s, a)$ to the Q-value, and it has the property of satisfying Bellman equation, so the temporal difference (TD) (Tesauro, 1995) is generally used to minimize Bellman errors by the transition tuple $(s, a, r, s')$ at every critic evaluation step, which is given by $\mathbb{E}_{(s, a, r, s')} \left[ (r + \gamma Q^t(s', \pi(s')) - Q(s, a))^2 \right]$ (Lillicrap et al., 2015), where $Q^t$ stands for a target Q network. In algorithms using the experience replay, $(s, a, r, s')$ will be stored in a replay buffer at every environment step, $a$ is sampled from the experience pool, and the next action has to be judged by the current policy, represented as $\pi(s')$. In off-policy methods, the distribution of sampled action $a$ is different from the current policy, which will cause distribution mismatch. In the context, we use the term of 'iteration' to represent the index of updates. In the actor-critic paradigm, each iteration contains the evaluation step and the policy improvement step, which are used to update Q-value function and optimize the policy, respectively. After minimizing the Bellman errors, the policy improvement is performed by maximizing the expected return $J(\theta) = \mathbb{E}_s [Q_\pi(s, \pi(s)]$. In some algorithms, the policy regularization may be attached to the expected return for the stability of training (Kumar et al., 2019; Jaques et al., 2019), which is aimed to restrain the policy gradient $\nabla_\theta J(\theta)$ to keep away from potential gradient vanishing or exploding problems as well as reducing the estimation variance.

## 4 MODEL-BASED ACTOR-CRITIC VALUE EXPLORATION WITH ASYMPTOTIC PLANNING

The actor-critic target value exploration with asymptotic planning is a method that blends the one-step global model-based prediction into target critic value, whose importance lies in how and why it

can work well. The diversity of state distributions between model-based prediction and observations produces extra value exploration, and asymptotically accurate learned model has the potential to overcome errors from the overestimation and mismatched policy distributions, when experience replay is applied in off-policy algorithms. The asymptotic planning is realized by gradually increasing the weight of model-based prediction in the target critic value.

## 4.1 MODEL-BASED TARGET VALUE EXPLORATION

By instinct, the model-based target value is able to explore the future states from different viewpoints with some certainty. Without the model-based prediction, the target value usually chooses the input next states from random samples in replay buffer. The distributions of sampled next states which meet $s_{t+1} \sim P^{t+1}(s_{t+1})$ (2) are not really stationary in off-policy methods, since the policy will go through multiple updates in a rollout. These unstationary distributions can distort the real stationary transition probability from the view of the current policy, which means a false dynamics could be experienced or "felt" with the real observations from the replay buffer, which are achieved by interacting with the environment. The problem of distribution mismatch will be incurred when off-policy method is used, since actions $(a_t, \cdots, a_0)$ are random samples from the replay buffer following different distributions. The analyses tell that observations from the real environment are not necessarily more accurate than the prophecy of a dynamics model.

Based on the choice of target value, the evaluation step for the critic updates can be separated into the off-policy training and the on-policy planning. We introduce the backup operator for model-based Q-value prediction as

$$\mathcal{P}^\pi Q(s_t, a_t) = r_\mu(s_t, a_t, s_{t+1}^p) + \gamma \mathbb{E}_{s_{t+1}} \left[ Q(s_{t+1}^p, a_{t+1}) \right], \tag{5}$$

where $s_{t+1}^p \sim p(\cdot|s_t, a_t)$, $a_t \sim \pi(\cdot|s_t)$, $a_{t+1} \sim \pi(\cdot|s_{t+1})$, and $r_\mu$ and $p$ are reward functions and transition probability learnt from the dynamics model, respectively. We use $s_{t+1}^p$ to distinguish the subtle difference between the model-based prediction and $s_{t+1} \sim P^{t+1}(s_{t+1})$, which follows a distribution determined by the complete rollout. And $a_{t+1} \sim \pi(\cdot|s_{t+1})$ because of the limitation on one-step global prediction, otherwise, choosing $a_{t+1} \sim \pi(\cdot|s_{t+1}^p)$ will induce cumulative model bias. If replacing $r_\mu$ and $s_{t+1}^p$ with the immediate reward $r$ and the next state $s'$ from past experience, (19) reduces to the target Q-value of deep deterministic policy gradient (DDPG) (Lillicrap et al., 2015). Plus a regularization term concerning the entropy policy exploration, it then becomes the target Q-value of SAC.

The change of target value seems subtle, however, the model-based prediction produces valid state and reward diversity for value exploration. Unlike the action which is usually bounded, the state space is continuously unbounded in many tasks like Gym environments, then the exploration strategy commonly used in policy exploration, for example the gaussian exploration noise, will be invalid in value exploration. Throughout the referred literature, few works have attempted to apply simple and/or feasible value exploration. More importantly, as the trained reward model and transition model grow more accurate, the on-policy prediction (19) will greatly reduce or avoid originally existing overestimation errors and distribution mismatch in target Q-value without value exploration. Although model bias is induced by model-based planning during learning, it can be controlled by asymptotically increased impact of model-based prediction.

**Lemma 2.** *Consider the sequence $Q_{t+1} = \mathcal{P}^\pi Q_t$ constructed by (19), then given the condition that the Q-values are bounded, i.e., $|Q_t(s, a)| < \infty$, $\forall (s, a) \in \mathcal{S} \times \mathcal{A}$, the sequence $Q_t$ will converge to a unique optimal value as $t \to \infty$.*

The proof of Lemma 5 can be found in Appendix B. In this work, (19) will be combined with the target Q-value of SAC according to an asymptotically increasing percentage. To expand the value exploration, the current Q-value function to be predicted by the target Q-value is also divided into two parts sharing the same critic network parameter. However, they take actions following different distributions as inputs. By this means, the diversity can be enlarged and some convergence conditions can be satisfied, which will be shown later in Theorem 4.

In the policy improvement step, we directly adopt the entropy policy exploration of SAC with the actor network, which does not involve computation on the dynamics. When the model-free DRL is applied to large-scale continuous control problems, the dynamics is unknown and the state-action

spaces are continuous, the policy improvement over $\mathcal{S} \times \mathcal{A}$ at every iteration, which is called as the absolute policy improvement, cannot be guaranteed by estimation with distribution mismatch and estimation biases. Some researches on continuous control also show empirical results which degrade after reaching a good point. As told in related work part, there have been several methods trying to apply the model-based synthetic rollouts to the policy improvement step, however, they do not work well with the proposed value exploration strategy according to our practice since the one-step global prediction will be violated.

## 4.2 DYNAMICS LEARNING

Compared with descriptive models that are feasible in small state spaces (Deisenroth & Rasmussen, 2011; Khansari-Zadeh & Billard, 2011), neural network approximation can scale better to high-dimensional state spaces. We parameterize the dynamics $p_\lambda(s, a)$ and $r_\mu(s, a, s')$ with deep generative models (Moerland et al., 2020), where the parameters $\lambda$ and $\mu$ reparameterize the transition and the reward functions, respectively. Considering the fact that the transition function is difficult to train, we choose to learn a relative transition function to forecast the difference between the current state and next state, which is given by $s'_p = s + p_\lambda(s, a)$, and the state difference follows the relative transition probability density function (pdf) $p_\lambda(\cdot|s, a)$. The use of $s$ in does not induce the distribution mismatch since the relative transition function is unaffected by the policy. The representation of unknown reward function differs in the inputs among various tasks, for example, the information determining the rewards is not included in observations for the default setting of MuJoCo suite (Todorov et al., 2012; Brockman et al., 2016). This complicates the training of the reward function, but we will show that taking inputs as $(s, a, s')$ is applicable in our selected benchmarks.

**Lemma 3.** *Assume the absolute value of expected reward function and the expected KL-divergence between the dynamics model and the real transition probability are respectively bounded by*

$$
\max_{s \sim \mathcal{S}} |\mathbb{E}_{a \sim \pi} r(s, a)| \leq r_m,
$$
$$
\max_t \mathbb{E}_{a_t \sim \pi, s_t \sim P^t} D_{TV}(p(s_{t+1}|s_t, a_t) || P(s_{t+1}|s_t, a_t)) \leq \delta, \tag{6}
$$

*then we have*

$$
\left| \mathbb{E}_{s_t^p \sim p^t, a_t \sim \pi} [Q(s_t^p, a_t)] - \mathbb{E}_{s_t \sim P^t, a_t \sim \pi} [Q(s_t, a_t)] \right| \leq O(\delta). \tag{7}
$$

The proof of Lemma 6 can be found in Appendix C, which tells the distance between predicted Q-value and true Q-value is bounded linearly by $\delta$. The dynamics Model is trained accompanying the iteration, using random samples from experience buffer. Given the four-tuple sample $(s, a, r, s')$, the surrogate objective of relative transition function is given by

$$
D(\lambda) = \mathbb{E}_{(s, a, s') \sim \mathcal{R}} \left[ \frac{1}{2}(s + p_\lambda(s, a) - s')^2 \right], \tag{8}
$$

where $\mathcal{R}$ represents the replay buffer that random samples come from.

Then (8) can be optimized with stochastic gradient

$$
\hat{\nabla}_\lambda D(\lambda) = \mathbb{E}_{(s, a, s')} \left[ (s + p_\lambda(s, a) - s') \hat{\nabla}_\lambda p_\lambda(s, a) \right]. \tag{9}
$$

Similarly, the surrogate objective for updating the reward function can be formulated as

$$
D(\mu) = \mathbb{E}_{(s, a, r, s') \sim \mathcal{R}} \left[ \frac{1}{2}(r_\mu(s, a, s') - r)^2 \right], \tag{10}
$$

and it gradient is computed as

$$
\hat{\nabla}_\mu D(\mu) = \mathbb{E}_{(s, a, r, s')} \left[ (r_\mu(s, a, s') - r) \hat{\nabla}_\mu r_\mu(s, a, s') \right]. \tag{11}
$$

A well trained reward function is necessary for (19). Without the real-time model of reward function, sampled rewards from the replay buffer will induce distribution errors in model-based prediction.

### 4.3 MODEL-BASED ACTOR-CRITIC VALUE EXPLORATION ALGORITHM

As mentioned above, we adopt the minimized pairwise critics to serve as the target Q-value for the purpose of mitigating the effect of overestimation (Watkins, 1989). Besides, current and target networks are separated to execute soft updates (Lillicrap et al., 2015; Haarnoja et al., 2018b) for all surrogate objectives, for the good of stability. In this work, the asymptotical model-based prediction based on (19) is merged into the target value of SAC, so the loss function for the update of critic parameters in the evaluation step can be estimated by

$$L(\omega^i) = \mathbb{E}_{(s,a,r,s')}\left[(kQ^t(s_p', a') + (1-k)Q^t(s', a') - kQ_{\omega^i}(s, \overline{a}) - (1-k)Q_{\omega^i}(s, a))^2\right], \quad (12)$$

where $i \in \{1, 2\}$, $\overline{a} = \pi_\theta(s)$ and $a' = \pi_{\theta'}(s')$ are the on-policy actions chosen from the current policy and the target policy parameterized by $\theta$ and $\theta'$, respectively, $(s, a, r, s')$ is a tuple of history data sampled from the experience pool, and $k$ is the asymptotic variable increasing from 0 to 1 as the time step proceeds. And

$$Q^t(s', a') = r + \gamma\left[\min\left(Q_{\omega'^1}(s', a'), Q_{\omega'^2}(s', a')\right) - \frac{\alpha}{1-k}\log(\pi_{\theta'}(a'|s'))\right], \quad (13)$$

$$Q^t(s_p', a') = \gamma\min\left(Q_{\omega'^1}(s_p', a'), Q_{\omega'^2}(s_p', a')\right) + r_\mu(s, \overline{a}, s_p'), \quad (14)$$

where $\omega^1$, $\omega^2$, $\omega'^1$ and $\omega'^2$ parameterize two critic networks and their target estimates, respectively. Besides, $s_p' = s + p_\lambda(s, \overline{a})$ is the on-policy next state predicted by the transition model parameterized by $\lambda$, and $\pi_{\theta'}(\cdot|s')$ is the target policy distribution conditioned on the next state $s'$. The action input of reward function $r_\mu$ follows the current policy instead of being sampled from the replay buffer. By minimizing (24), the critic parameters can be updated for each evaluation step.

**Theorem 1.** *Assume* $\mathbb{E}_{(s,a,r,s')}\left[(Q^t(s', a') - Q_{\omega^i}(s, a))^2\right] \leq \epsilon$ *for* $t > T_1$*, then* $\exists T > 0$ *so that* $L(\omega^i) \leq 2\epsilon$ *for* $t > T$.

The proof of Theorem 3 can be found in Appendix D. Theorem 3 means that the critic loss function (24) of this work will converge under the assumption that the critic loss of SAC converges. Besides the conclusion of convergence, we are more curious about how it converges or by what factors it is affected. From Theorem 4, we see both the accuracy of transition model and reward function model will affect the target value prediction, and the transition model influences more. Moreover, a well-trained model can guarantee its convergence.

**Theorem 2.** *Assume the absolute value of expected reward function, the expected KL-divergence between the dynamics model and the real transition probability, and the MSE of expected difference between the modeled reward function and the immediate reward are respectively bounded by*

$$\max_{s \sim \mathcal{S}} |\mathbb{E}_{a \sim \pi} r(s, a)| \leq r_m,$$

$$\max_t \mathbb{E}_{a_t \sim \pi, s_t \sim P^t} D_{TV}(p(s_{t+1}|s_t, a_t)||P(s_{t+1}|s_t, a_t)) \leq \delta,$$

$$\max_t \mathbb{E}_{(s,r)}\left\{\mathbb{E}_{s_{t+1}^p \sim p^{t+1}, \overline{a}_t \sim \pi}\left[r_\mu(s, \overline{a}_t, s_{t+1}^p) - r\right]\right\}^2 \leq \xi, \quad (15)$$

*then we have the MSE of target prediction error bounded by* $2\xi + O(\delta^2)$.

*Proof*  See Appendix E. This theorem interprets that the target prediction error does not suffer from overestimation bias and mismatched distribution, and can be negligible once the learnt model is accurate, then the prediction error distance is bounded by $\sqrt{2\xi + O(\delta^2)}$. $\qquad\square$

Generally, the policy improvement step aims to maximize the current Q-value or it variant, which does not need the dynamics model to predict the reward or the next state. Then the surrogate objective function used to update the actor parameters can be directly given by

$$J(\theta) = \mathbb{E}_s\left[Q_{\omega^{1,2}}(s, a) - \alpha\log(\pi_\theta(a|s))\right], \quad (16)$$

where $Q_{\omega^{1,2}}(s, a) = \min\left(Q_{\omega^1}(s, a), Q_{\omega^2}(s, a)\right)$, $s$ comes from the tuple of history data, $a = \pi_\theta(s)$ is the reparameterized action based on $s$ and the policy network parameterized by $\theta$, and $\pi_\theta(\cdot|s)$ is the current policy distribution conditioned on the current state $s$. By maximizing (16), the actor parameter can be updated for each policy improvement step.

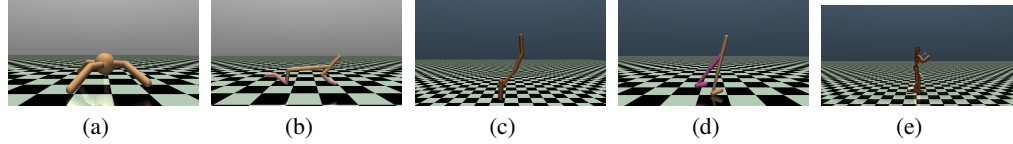

Figure 1: (a) Ant-v3; (b) Halfcheetah-v3; (c) Hopper-v3; (d) Walker2d-v3; (e) Humanoid-v3

For updates of target critic parameters, we adopt "soft" target updates (Lillicrap et al., 2015) using a weighted factor $0 \leq \tau < 1$ to control the speed of policy updates for the sake of small value error at each iteration. Except the critic parameters, we adopt immediate updates for other parameters in this work. In (9) and (11), the gradients in expectation forms are approximated by averaging over the sampled results of rollouts from past experience. We organize the above procedures as the model-based actor-critic value exploration with asymptotic planning (MAVE) algorithm, whose pseudocode is described by Algorithm 1. The algorithm alternates between running the environment steps to collect experience, and updating the network parameters using the stochastic gradients computed by the sampled batches from the experience pool. It is composed of online training and off-line planning, separated by the maximum online time step $T_1$. At the stage of online training, Step 10 in Algorithm 1 requires tiny computation using the dynamics model without the burden of great computing and storage processing from virtual or synthetic rollouts, as explained by (14).

---

**Algorithm 1** MAVE Algorithm

1: Initialize parameters $\omega^1 \leftarrow \omega_0^1$, $\omega^2 \leftarrow \omega_0^2$, $\theta \leftarrow \theta_0$, $\lambda \leftarrow \lambda_0$, $\mu \leftarrow \mu_0$
2: Initialize target parameters $\omega'^1 \leftarrow \omega_0'^1$, $\omega'^2 \leftarrow \omega_0'^2$, $\theta' \leftarrow \theta_0'$
3: Initialize the learning rates $l_c, l_a, l_d$ for the critic, the actor and the dynamics model, the time step $t \leftarrow 0$, the asymptotic variable $k \leftarrow 0$, the soft update hyperparameter $\tau$, the maximum online time step $T_1$, the maximum overall time step $T$, the batch size $B$ and the replay buffer $\mathcal{R} \leftarrow \emptyset$.
4: **while** $t < T_1$ **do**
5:      Select action $a_t \sim \pi_{\theta_t}(a_t|s_t)$
6:      Observe the reward and next state from the interaction feedback
7:      Store transition $\mathcal{R} \leftarrow \mathcal{R} \cup \{(s_t, a_t, r_t, s_{t+1})\}$
8:      Sample a batch of transitions $\mathcal{B} = (s, a, r, s')_{i=1}^B$ from $\mathcal{R}$
9:      **for** each time step **do**
10:          $\omega_{t+1}^i \leftarrow \omega_t^i - l_c \hat{\nabla}_{\omega_t^i} L(\omega_t^i)$ for $i \in \{1, 2\}$ following $\hat{\nabla}_{\omega^i} L(\omega^i)$
11:          $\theta_{t+1} \leftarrow \theta_t + l_a \hat{\nabla}_{\theta_t} J(\theta_t)$ following $\hat{\nabla}_{\theta^i} J(\theta^i)$
12:          $\lambda_{t+1} \leftarrow \lambda_t - l_d \hat{\nabla}_\lambda D(\lambda)$ following Eq. (9)
13:          $\mu_{t+1} \leftarrow \mu_t - l_d \hat{\nabla}_\mu D(\mu)$ following Eq. (11)
14:          $\omega'^i_{t+1} \leftarrow \tau \omega_{t+1}^i + (1 - \tau)\omega'^i_t$ for $i \in \{1, 2\}$
15:      **end for**
16:      $s_{t+1} \leftarrow s_t$;
17:      $t \leftarrow t + 1$
18: **end while**
19: $k \leftarrow 1$
20: **while** $t < T$ **do**
21:      Repeat Steps 10, 11, 12, 13, 14 and 17
22: **end while**

---

## 5 EXPERIMENTS

### 5.1 BENCHMARKS

The performance of our proposed method is compared with several prior model-free and model-based reinforcement learning algorithms in the sample complexity and stability on a set of gym

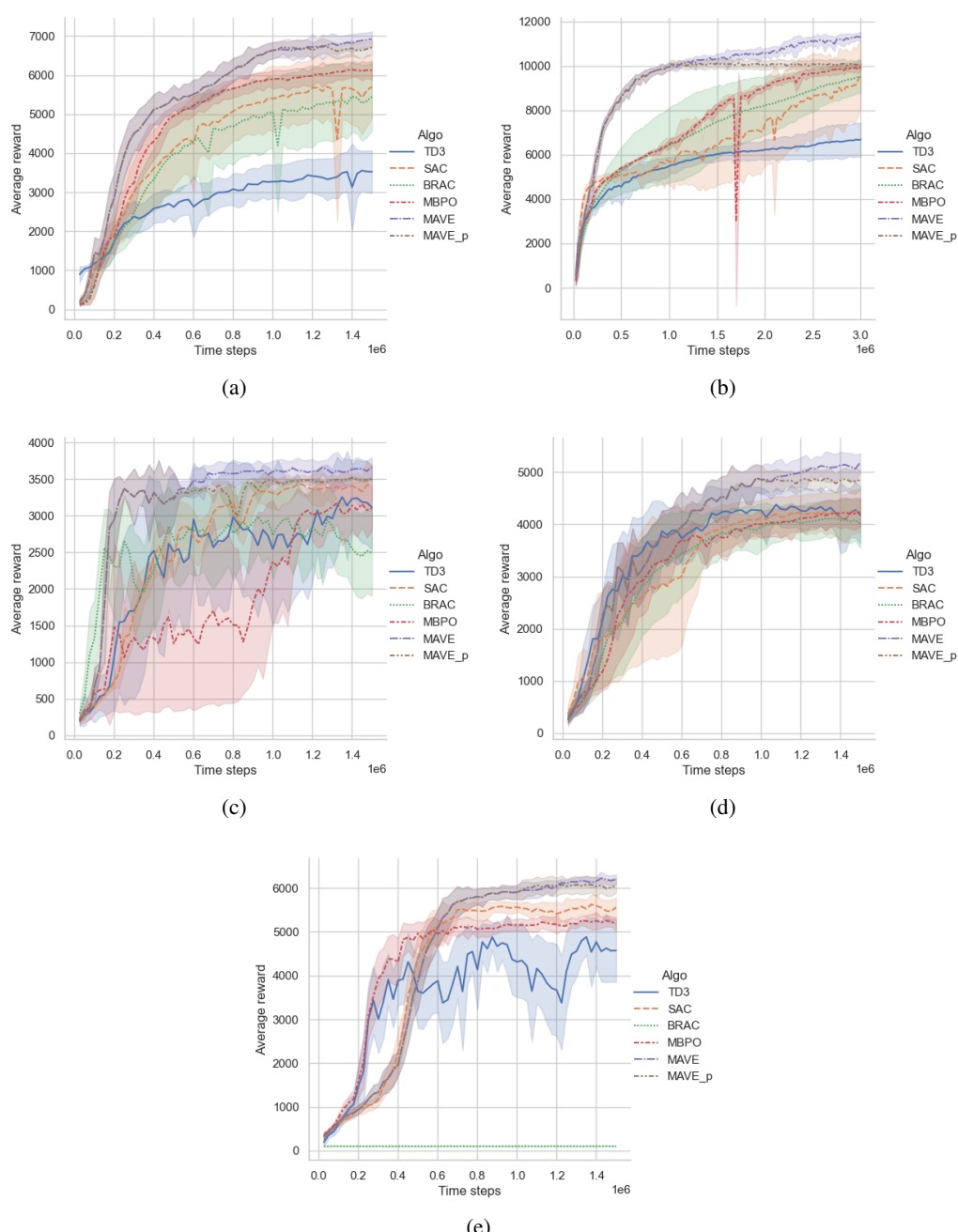

Figure 2: Average reward of off-line training and online-planning versus time step in (a) Ant-v3; (b) Halfcheetah-v3; (c) Hopper-v3; (d) Walker2d-v3; (e) Humanoid-v3

continuous control tasks, several of which are selected in this paper as benchmarks illustrated in Fig. 1.

## 6 BASELINES

The baselines adopted for reference includes TD3, SAC, BRAC and Model-based policy optimization (MBPO) (Janner et al., 2019). Before the appearance of SAC, DDPG is regarded as one of the most efficient off-policy DRL methods (Duan et al., 2016), followed by TD3 as an extension. SAC

has achieved state-of-the-art sample efficiency in multiple challenging continuous control domains (Christodoulou, 2019), and BRAC can be regarded as a variant of SAC by adopting an extra policy regularization based on the KL divergence between updated and older policy. In this work, we adopt (8) to train the transition probability instead of maximum likelihood since the logarithm of transition probability parameterized by the global gaussian network tends to be unbounded.

We apply the shared hyperparameters to our proposed algorithm with other baselines for every benchmark to keep fairness. In the process of collecting the off-policy rollouts, the gaussian exploration noise is added to every time step with a fixed variance 0.2 when choosing the action, and then the noisy action is clipped within the set boundary to avoid out-of-distribution (OOD) actions (Kumar et al., 2019; 2020). The discount horizon factor is selected as 0.99, and all algorithms adopt stochastic policies and maximum prior action entropy except for TD3. The stochastic policies follow gaussian distributions with mean and variance parameterized by fully connected networks with two hidden layers, each of which has 256 units. TD3 uses a deterministic policy, also parameterized by fully connected networks with two hidden layers. We organize the network architectures and hyperparameters in Appendix F and G, respectively. The Adam optimizer (Kingma & Ba, 2014) is used to update the network parameters.

## 6.1 RESULTS

We run 10 seeds numbered from 0 to 9 for each algorithm to keep a fair comparison. After every 500 iterations (time steps), we launch a evaluation procedure, which averages 10 rollouts for a test. The average reward of a test will be recorded at every evaluation procedure, and all tests throughout the time step scale give the result of each algorithm.

The average rewards of algorithms tested in chosen benchmarks are shown Fig. 2 with standard deviation as the confidence interval (CI). From Figs. 2(a), 2(b) and 2(c), we can observe higher converged value and smaller standard deviation of MAVE at late time steps over other baselines. At early stage before 1 million time steps, MAVE vibrates due to the training dynamics, as we note from Figs. 2(a), 2(c) and 2(d). In Hopper environment, since the converged value is far lower than other benchmarks, the tolerance for the fluctuation around convergence is much lower, which causes the instability problems of tested baselines. However, MAVE shows strong robustness and has a converged value up to 3700 compared with other baselines, which outweighs the limits of state-of-the-art results in Hopper task, as shown in 2(c). In Fig. 2(d), MAVE has a relatively stable performance over 5000. For Humanoid with high-dimensional action space, Fig. 2(e) shows that MAVE is much better than other baselines and can converge around the score of 6200. Over all figures in Fig. 2, SAC and BRAC both have their up and downs, and TD3 gives the worst performance, considering its lack of adopting the policy exploration.

We also show the goodness of trained dynamics model by plotting the results of off-line learning without interacting with the environment after $T_1$ in Fig. 2, labeled by 'MAVE-P', which is analyzed in Appendix H.

By the way, due to the differences in code details and the Mujoco version (which we use version 3), the converged maximum may be a bit different from those of (Haarnoja et al., 2018b) and (Feinberg et al., 2018). For example, the best result of Halfcheetah is 8000 in (Feinberg et al., 2018), however, it is up to 15000, which is 12000 in our case. In contrast, the best result of Ant is 6000 in (Haarnoja et al., 2018b), which is lower than 7000 in our work and similar to Janner et al. (2019). Except for Ant, the best results in Janner et al. (2019) is closer to ours'.

Due to the page limit, the ablation studies can be found in Appendix I.

## 7 CONCLUSION

In this paper, we proposed a method that combines notations of dynamics training, model prediction, off-line training and on-line planning, which jointly deduce a simple solution to the value exploration. Our work is sensitive to the dynamics precision, especially to the transition model, however, it is free from costs on extra storage and computation and can greatly reduce estimation errors and distribution mismatches. The minor costs of our work are the necessary networks for the dynamics and a hyperparameter (given in Appendix G) to control the speed of asymptotical dynamics training.

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

## A  PROOF OF LEMMA 4

**Lemma 4.** *Assume the expected KL-divergence between two transition distributions is bounded by*

$$\max_t \mathbb{E}_{a_t \sim \pi, s_t \sim P^t} D_{TV}(p(s_{t+1}|s_t, a_t)||P(s_{t+1}|s_t, a_t)) \leq \delta, \tag{17}$$

*then we have* $\mathbb{E}_{s_t \sim \mathcal{S}} |p^t(s_t) - P^t(s_t)| \leq 2t\delta$.

*Proof*

$$
\begin{aligned}
&\mathbb{E}_{s_t \sim \mathcal{S}} \left| p^t - P^t \right| \\
=& \mathbb{E}_{s_t \sim \mathcal{S}, s_{t-1} \sim \mathcal{S}, a_{t-1} \sim \pi} \left| p(s_t|s_{t-1}, a_{t-1}) p^{t-1} - P(s_t|s_{t-1}, a_{t-1}) P^{t-1} \right| \\
\leq& \mathbb{E}_{s_t \sim \mathcal{S}, s_{t-1} \sim \mathcal{S}, a_{t-1} \sim \pi} \left| p(s_t|s_{t-1}, a_{t-1})[p^{t-1} - P^{t-1}] + P^{t-1}[p(s_t|s_{t-1}, a_{t-1}) - P(s_t|s_{t-1}, a_{t-1})] \right| \\
\leq& \mathbb{E}_{s_{t-1} \sim \mathcal{S}, a_{t-1} \sim \pi} \left| p^{t-1} - P^{t-1} \right| + \mathbb{E}_{s_t \sim \mathcal{S}, s_{t-1} \sim P^{t-1}, a_{t-1} \sim \pi} \left| p(s_t|s_{t-1}, a_{t-1}) - P(s_t|s_{t-1}, a_{t-1}) \right| \\
=& \mathbb{E}_{s_{t-1} \sim \mathcal{S}} \left| p^{t-1} - P^{t-1} \right| + 2\mathbb{E}_{a_{t-1} \sim \pi, s_{t-1} \sim P^{t-1}} D_{TV}(p(s_t|s_{t-1}, a_{t-1})||P(s_t|s_{t-1}, a_{t-1})) \\
\leq& \mathbb{E}_{s_{t-1} \sim \mathcal{S}} \left| p^{t-1} - P^{t-1} \right| + 2\delta \\
\leq& \left| p(s_0) - P(s_0) \right| + 2t\delta \\
=& 2t\delta, \tag{18}
\end{aligned}
$$

where $p^t = p^t(s_t)$ and $P^t = P^t(s_t)$, and the last equality holds because the initial distribution is not affected by transitions. This proof is partly referred to Janner et al. (2019). □

# B   PROOF OF LEMMA 5

$$\mathcal{P}^\pi Q(s_t, a_t) = r_\mu(s_t, a_t, s_{t+1}^p) + \gamma \mathbb{E}_{s_{t+1}} \left[ Q(s_{t+1}^p, a_{t+1}) \right], \tag{19}$$

**Lemma 5.** *Consider the sequence $Q_{t+1} = \mathcal{P}^\pi Q_t$ constructed by (19), then given the condition that the Q-values are bounded, i.e., $|Q_t(s,a)| < \infty$, $\forall (s,a) \in \mathcal{S} \times \mathcal{A}$, the sequence $Q_t$ will converge to a unique optimal value as $t \to \infty$.*

*Proof*

$$
\begin{aligned}
&|\mathcal{P}^\pi Q(s_t, a_t) - \mathcal{P}^\pi Q'(s_t, a_t)| \\
\leq & \gamma \left| \mathbb{E}_{s_{t+1}} [Q(s_{t+1}^p, a_{t+1}) - Q'(s_{t+1}^p, a_{t+1})] \right| \\
\leq & \gamma \max_{s_{t+1}} \left| Q(s_{t+1}^p, a_{t+1}) - Q'(s_{t+1}^p, a_{t+1}) \right| \\
= & \gamma \| Q - Q' \|_\infty,
\end{aligned}
\tag{20}
$$

where $\|\cdot\|_\infty$ means the max norm. Since the Q-value is assumed to be bounded, the second inequality holds. We reach a conclusion that $\forall (s_t, a_t) \in \mathcal{S} \times \mathcal{A}$, (20) holds, which can be rewritten as $\|T^\pi Q - T^\pi Q'\|_\infty \leq \gamma \| Q - Q' \|_\infty$, which means $P^\pi Q(s, a)$ is a max-norm contraction mapping. According to the contraction property, the sequence $Q_{k+1} = \mathcal{P}^\pi Q_k$ will converge to a unique fixed point. $\square$

# C   PROOF OF LEMMA 6

**Lemma 6.** *Assume the absolute value of expected reward function and the expected KL-divergence between the dynamics model and the real transition probability are respectively bounded by*

$$\max_{s \sim \mathcal{S}} |\mathbb{E}_{a \sim \pi} r(s, a)| \leq r_m,$$
$$\max_t \mathbb{E}_{a_t \sim \pi, s_t \sim P^t} D_{TV}(p(s_{t+1}|s_t, a_t) || P(s_{t+1}|s_t, a_t)) \leq \delta, \tag{21}$$

*then we have*

$$\left| \mathbb{E}_{s_t^p \sim p^t, a_t \sim \pi} \left[ Q(s_t^p, a_t) \right] - \mathbb{E}_{s_t \sim P^t, a_t \sim \pi} \left[ Q(s_t, a_t) \right] \right| \leq O(\delta). \tag{22}$$

*Proof*

$$
\begin{aligned}
& \left| \mathbb{E}_{s_t^p \sim p^t, a_t \sim \pi} \left[ Q(s_t^p, a_t) \right] - \mathbb{E}_{s_t \sim P^t, a_t \sim \pi} \left[ Q(s_t, a_t) \right] \right| \\
= & \left| \sum_t \mathbb{E}_{s_t \sim \mathcal{S}, a_t \sim \pi} \left[ \gamma^t r(s_t, a_t)(p^t(s_t) - P^t(s_t)) \right] \right| \\
\leq & \sum_t \mathbb{E}_{s_t \sim \mathcal{S}} \left| \mathbb{E}_{a_t \sim \pi} [r(s_t, a_t)] \gamma^t (p^t(s_t) - P^t(s_t)) \right| \\
\leq & r_m \sum_t \mathbb{E}_{s_t \sim \mathcal{S}} \left[ \gamma^t \left| p^t(s_t) - P^t(s_t) \right| \right] \\
\leq & 2 r_m \delta \sum_{i=0}^t [i \gamma^i] \\
\leq & \frac{2 r_m \delta \gamma}{(1 - \gamma)^2},
\end{aligned}
\tag{23}
$$

where the second last inequality is due to Lemma 4. $\square$

# D   PROOF OF THEOREM 3

**Theorem 3.** *Assume $\mathbb{E}_{(s,a,r,s')} \left[ (Q^t(s', a') - Q_{\omega^i}(s, a))^2 \right] \leq \epsilon$ for $t > T_1$, then $\exists T$ so that $L(\omega^i) \leq 2\epsilon$ for $t > T$.*

*Proof*    According to Lemma 5, $\exists T_2 > 0$, $\mathbb{E}_{(s,a,r,s')}\left[(Q^t(s',a') - Q_{\omega^i}(s,a))^2\right] \leq \epsilon$ for $t > T_2$, then for $T > \max T_1, T_2$

$$L(\omega^i) \leq 2\mathbb{E}_{(s,a,r,s')}\left[k^2(Q^t(s'_p,a') - Q_{\omega^i}(s,\overline{a}))^2 + (1-k)^2(Q^t(s',a') - Q_{\omega^i}(s,a))^2\right],$$

$$\leq 2\epsilon, \tag{24}$$

for $t > T$.  □

## E    PROOF OF THEOREM 4

**Theorem 4.** *Assume the absolute value of expected reward function, the expected KL-divergence between the dynamics model and the real transition probability, and the MSE of expected difference between the modeled reward function and the immediate reward are respectively bounded by*

$$\max_{s \sim \mathcal{S}} |\mathbb{E}_{a \sim \pi} r(s,a)| \leq r_m,$$

$$\max_t \mathbb{E}_{a_t \sim \pi, s_t \sim P^t} D_{TV}(p(s_{t+1}|s_t, a_t)||P(s_{t+1}|s_t, a_t)) \leq \delta,$$

$$\max_t \mathbb{E}_{(s,r)} \left\{ \mathbb{E}_{s^p_{t+1} \sim p^{t+1}, \overline{a}_t \sim \pi} \left[ r_\mu(s, \overline{a}_t, s^p_{t+1}) - r \right] \right\}^2 \leq \xi, \tag{25}$$

*then we have the MSE of target prediction error bounded by $2\xi + O(\delta^2)$.*

*Proof*

$$\mathbb{E}_s \left\{ \mathbb{E}_{s^p_{t+1} \sim p^{t+1}, \overline{a}_t \sim \pi, a_{t+1} \sim \pi} \left[ r_\mu(s, \overline{a}_t, s^p_{t+1}) + \gamma Q(s^p_{t+1}, a_{t+1}) - Q(s, \overline{a}_t) \right] \right\}^2$$

$$= \mathbb{E}_{(s,r)} \left\{ \mathbb{E}_{s^p_{t+1} \sim p^{t+1}, s_{t+1} \sim P^{t+1}, \overline{a}_t \sim \pi, a_{t+1} \sim \pi} \left[ r_\mu - r + \gamma Q(s^p_{t+1}, a_{t+1}) - \gamma Q(s_{t+1}, a_{t+1}) \right] \right\}^2$$

$$\leq 2\xi + 2\gamma^2 \left\{ \mathbb{E}_{s^p_{t+1} \sim p^{t+1}, s_{t+1} \sim P^{t+1}, a_{t+1} \sim \pi} \left[ Q(s^p_{t+1}, a_{t+1}) - Q(s_{t+1}, a_{t+1}) \right] \right\}^2$$

$$\leq 2\xi + 2\gamma^2 \left\{ \mathbb{E}_{s^p_{t+1} \sim p^{t+1}, s_{t+1} \sim P^{t+1}, a_{t+1} \sim \pi} \left| Q(s^p_{t+1}, a_{t+1}) - Q(s_{t+1}, a_{t+1}) \right| \right\}^2$$

$$\leq 2\xi + \frac{8 r_m^2 \delta^2 \gamma^4}{(1-\gamma)^4}, \tag{26}$$

where $(s,r)$ is the random sample from the replay buffer, and the last inequality can be referred to Lemma 6. Then the prediction error is bounded by $\sqrt{2\xi + \frac{8 r_m^2 \delta^2 \gamma^4}{(1-\gamma)^4}}$.  □

## F    NETWORK ARCHITECTURE

We construct the critic network using a fully-connected MLP with two hidden layers. The input is composed of the state and action, outputting a value representing the Q-value. The ReLU functions are adopted to activate the two hidden layers. The setting of policy network follows normal random distribution, whose expectation and variance are fully-connected networks fed only by the state. Both of them have two hidden layers activated by the ReLU function. After the hidden layers, a Tanh function and a Softplus function follows to form the expectation and variance, respectively. With the expectation and variance, a normal distribution can be achieved to represent the random policy. The network of transition probability is constructed similarly to that of the policy without of the Tanh clipping, except for the input, which is composed of the state and action instead. And the network of reward function is similar to that of the critic, with the input composed of the state, action and the next state. The architecture of networks are plotted in Fig. 3. For simplicity, we omit the illustration of reward network in this figure. The above mentioned network architecture is adopted for the random policy. For the algorithm using the deterministic policy, the critic is constructed in the same way, however, the actor network is deterministic as the fully connected dense layer.

## G    HYPERPARAMETERS

Table 1 lists the common hyperparameters shared by all experiments and their respective settings. In this table, $L_a$ means the learning rate of the actor, $L_c$ means the learning rate of critics, and $L_d$ means

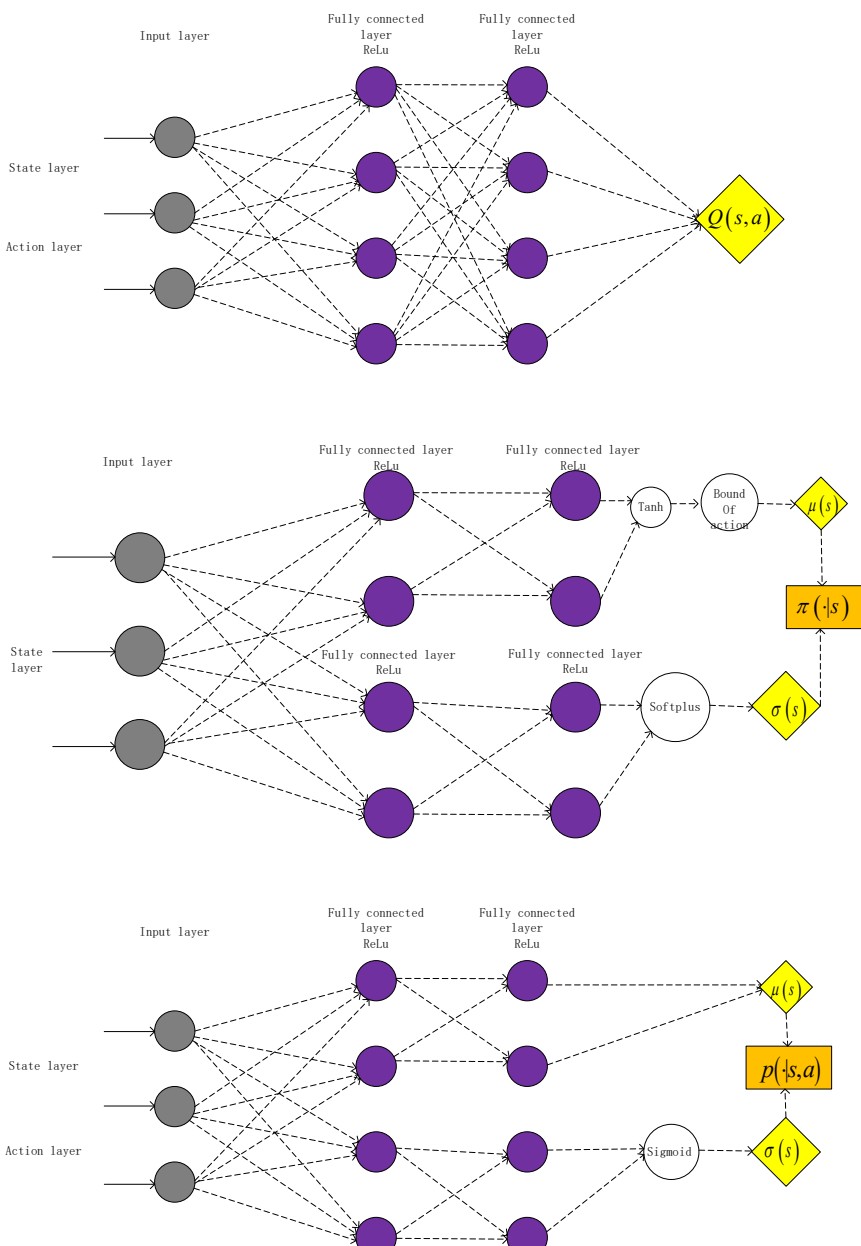

Figure 3: Architecture of networks.

the learning rate of dynamics including the transition probability and the reward. $\tau_a$ and $\tau_c$ represent soft update hyperparameter of the actor and the critic, respectively, and $\tau_a = 1$ means we adopt immediate update for the actor. The symbol $var$ represents the variance of gaussian exploration noise, and $\alpha$ is the fixed temperature hyperparameter for the term of maximum posteriori action entropy, which is applied in algorithms except DDPG and TD3. $\alpha_d$ represents the Wight factor of KL divergence for policy regularization applied in BRAC, $\beta$ is the temperature hyperparameter to tune the impact of posteriori transition entropy in MAVE, and $\eta$ is the asymptotic rise rate for $k_t = 1 - \beta^t$.

Table 1: **List of hyperparameters**

| Hyperparameter | Value | Description | Algorithm applied |
|---|---|---|---|
| $L_a$ | 0.0003 | Learning rate of actor | All |
| $L_c$ | 0.0003 | Learning rate of critic | All |
| $L_d$ | 0.0003 | Learning rate of dynamics | MAVE |
| $\tau_a$ | 1 | Soft update parameter of actor | All |
| $\tau_c$ | 0.005 | Soft update parameter of critic | All |
| $\gamma$ | 0.99 | Discount horizon factor | All |
| $var$ | 0.2 | The variance of exploration noise | All |
| $\alpha$ | 0.1 | Fixed temperature | Except DDPG and TD3 |
| $\beta$ | 0.1 | Fixed temperature | MAVE |
| $\alpha_d$ | 0.1 | Wight factor of KL regularization | BRAC |
| $\eta$ | 0.9999999995 | Asymptotic rise rate | MAVE |
| $Batch$ | 256 | Size of each mini-batch | All |
| $Units$ | 256 | Hidden layer units | All |
| $Memory$ | 1000000 | Size of replay buffer | All |
| $Interval$ | 500 | Evaluation period | All |
| $Test$ | 10 | Rollouts per evaluation | All |

Moreover, $Batch$ represents the size of mini-batches sampled for training, and $Memory$ is means the size of replay buffer. The rest in Table 1 are the hyperparameters for the evaluation procedure, specifically, $Interval$ means how many time steps between two successive evaluation procedures, and $Test$ means the number of rollouts run during each evaluation procedure.

## H    OFF-LINE PERFORMANCE

In Halfcheetah environment, the maximum online time step $T_1$ is set as 1 million, which means from 1 million to 3 million steps, the agent stops interacting with the environment and performs the training and planning totally based on the reserved fixed experience buffer, more specifically, based on the initial states $s$ in the experience four-tuple slots $(s, a, r, s')$. From Fig. 2(b), MAVE-P represents the off-line performance after $T_1$, which shows that the model can preserve the performance before the online training is stopped, when MAVE still has great potential to continue improving performance. Similar phenomena can be observed from Figs. 2(a), 2(d) and 2(e), with the same $T_1$ for Ant, Walker2d and Humanoid environments, respectively. The maximum online time step in Hopper is set as 0.5 million, which is smaller than other tasks because the upper converged value in Hopper is much lower. Besides, we note that Hopper does not reach a good point after stoping the on-line training, however, MAVE-P still manages to converge to 3500 solely counting on off-line training.

## I    ABLATION STUDY

To investigate the contribution of individual parts in the proposed value exploration, we replace the modeled reward function with the immediate reward, replace the model predicted next state $s'_p$ with buffer-sampled $s'$, and replace the on-policy current action $\bar{a}$ with buffer-sampled $a$ in Q-value function, labeled as 'MAVE-R', 'MAVE-P' and 'MAVE-Q', respectively. The figures of the three conditions are compared with MAVE in Fig. 4. From these figures, we see these fragments are all necessary. Specifically, lacking of the modeled reward function as part of the dynamics and the predicted next state foreseen by the modeled transition probability will induce distribution mismatches for MAVE-R and MAVE-P, respectively, and MAVE-Q will cause estimation error, which can be analyzed by the proof of Theorem 4.

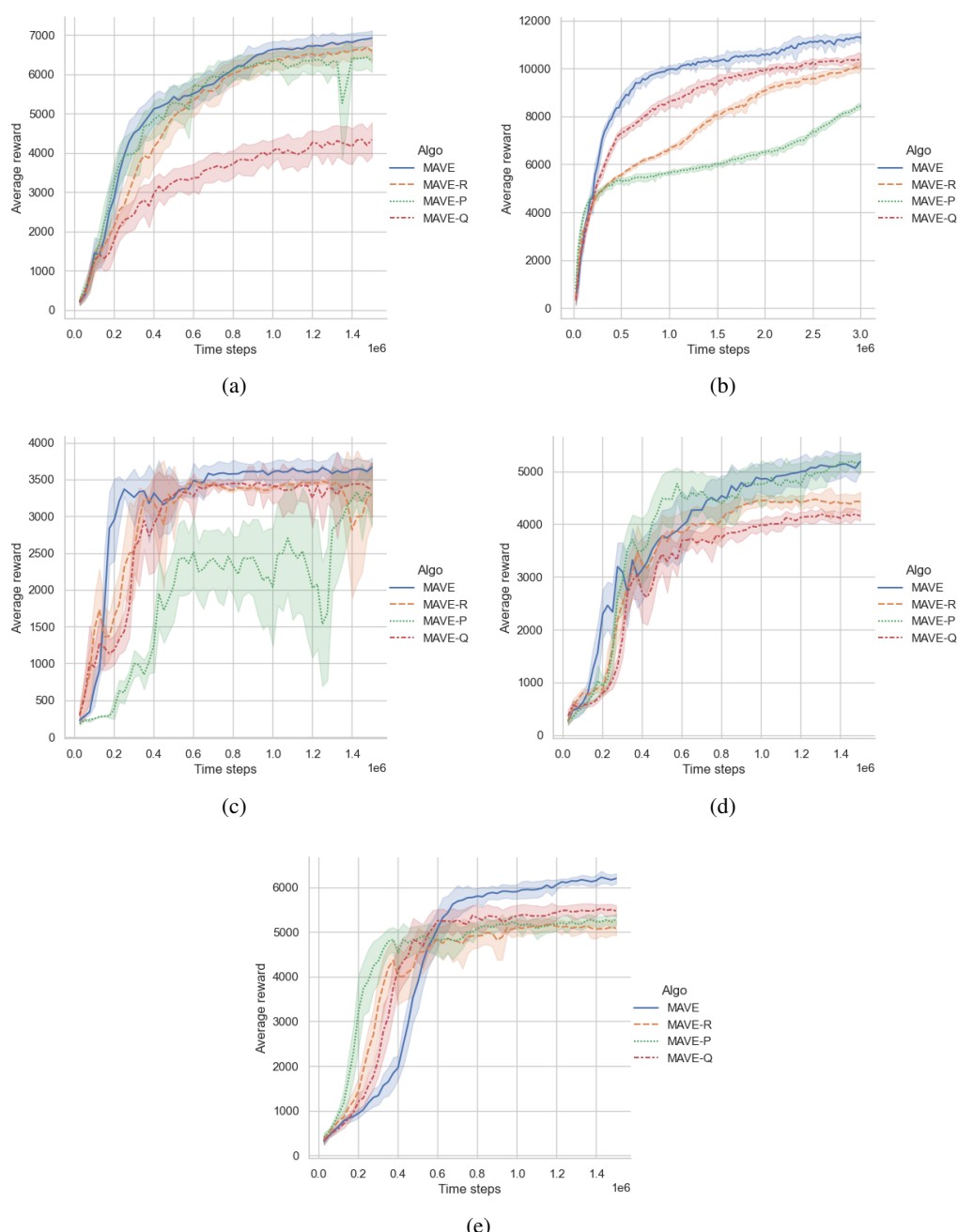

Figure 4: Ablation Study on 3 components in (a) Ant-v3; (b) Halfcheetah-v3; (c) Hopper-v3; (d) Walker2d-v3; (e) Humanoid-v3

