# OpenReview forum: "Model-based Value Exploration in Actor-critic Deep Reinforcement Learning"
_ICLR.cc/2023/Conference — Submitted to ICLR 2023_

### Official Review · Reviewer_YZy8 · 2022-10-23

**Confidence:** 4
**Clarity, Quality, Novelty And Reproducibility:** The overall writing is clear, with no…
**Correctness:** 2
**Technical Novelty And Significance:** 3
**Empirical Novelty And Significance:** 2
**Recommendation:** 3

**Strength And Weaknesses:**

Pros:

1. The motivation discussed in the paper is clear. The experiments show a clear performance boost in several tasks.

2.  The problem of interest, i.e. using MBRL to solve the distribution mismatch and over-estimation is a fruitful area.

Cons:

1. The claim that the proposed method can mitigate the issues in off-policy methods is weak. The authors only provide several standard theorems with very strong assumptions. Specifically, similar results are established in various MBRL works, e.g., MBPO [1]. The convergence results are not surprising based on the strong assumption: the model is accurate under every training policy. In other words, without the proper way of collecting diverse transition data, which requires exploration, claiming that model-based value target "provide extra exploration" is problematic. The conclusion that "convergence only depends on the accuracy of the learned model" is also crippled as it requires a globally accurate model.

2. As a result of Con 1, the claimed properties of the proposed method in the intro are not well supported by the theorem.

3. The contribution is limited, with connections to prior works not discussed. Can the authors discuss the connections with the model-based value expansion for model-free RL, e.g. by setting $H=1$ in MVE [2]? Since the reasons that the proposed method improves exploration and mitigates over-estimation are not clear, I am wondering if the method itself makes novel contributions.

4. The experiment shows performance improvement. But the underlying reasons why it benefits training are still not clear. The authors discuss several drawbacks of off-policy methods, so I would like to see experimental designs to demonstrate this (may in tabular MDPs) and how a learned model mitigates these issues.

[1] When to Trust Your Model: Model-Based Policy Optimization.

[2] Model-Based Value Expansion for Efficient Model-Free Reinforcement Learning

**Summary Of The Paper:**

This paper proposes to use dynamics model prediction as the value target value to mitigate several drawbacks of off-policy methods. The authors provide experiments and analysis in the model-based regime.

**Summary Of The Review:**

The idea itself is interesting, but the authors fail to establish either theoretical justification or experimental studies to support their claims. The existing results are based on strong assumptions, based on which the conclusion is questionable.

---

> ### Comment · Reviewer_YZy8 · 2022-11-17
> **No rebuttal response**
>
> Since the authors didn't engage in the rebuttal and the concerns regarding some claims are also held by other reviewers, I'm changing my score to 3.

---

### Official Review · Reviewer_Zou8 · 2022-10-23

**Confidence:** 4
**Correctness:** 3
**Technical Novelty And Significance:** 3
**Empirical Novelty And Significance:** 2
**Recommendation:** 3

**Clarity, Quality, Novelty And Reproducibility:**

* Presentation:
Unfortunately, the presentation of the paper is a mess and needs a major revision. From reading the paper once, it remains very unclear what the authors are doing, which is surprising as their idea is straightforward. The presentation feels a bit obfuscated to make the idea/contribution sound much bigger than it is. It would be great to simply say we mix off- and on-policy critic updates to achieve X. Furthermore, the authors sprinkle in many random theorems and lemma's to make the paper sound more theoretical/scientific. None of these lemmas/theorems have any importance for the algorithm and do not provide any insights. Therefore, these statements should be removed. Other points are:

- The numbering and references to equations, lemmas, and theorems are a mess. The paper constantly refers to Eq from the appendix. If you need to reference an Eq more than 5 times and is essential for the paper it should not be in the appendix. The numbering of theorems and lemmas is messed up as the numbers from the appendix are used and not the numbers introduced in the main part of the paper.

- Eq 12 should be rewritten such that it is obvious what is going on. I needed to stare at the Eq for a long time to figure it out. Something like  L = (k * TD_Q_off_policy + (1-k) * TD_Q_on_policy)**2 with adding in the entropy bonus would be great to make it more obvious.

- Fig. 2 needs cleaning up. It would be much better to consolidate the axis placement, and the legends and give each axis a title.

- "the Mujoco version (which we use version 3)" there is no MuJoCo 3.0, the current version is 2.3, which is only a few days old and not used in this paper.

- "Due to the page limit, the ablation studies can be found in Appendix I." The section on model learning is not useful as it is the standard way in the community. It could be halved and the gained space could be used for more important ablation studies.

**Strength And Weaknesses:**

* Technical Aspects:
The idea to mix off- & on-policy critic updates using a learned model is sound but also not super innovative. It is good that somebody tried it and reported the differences compared to the off-policy critic updates. Most design decisions are technically correct. Only the decision to evaluate the Q-function at t+1 with the predicted state of the dynamics model, i.e. s^p_{t+1}, with the action of the observed state, i.e., a_{t+1} \sim p(s_{t+1}) seems weird. I do not fully understand why one would evaluate a Q-function with an action that would not have been selected in this state. The argument:

> And a_t+1 ∼ π(·|s_t+1) because of the limitation on one-step global prediction, otherwise, choosing a_t+1 ∼ π(·|sp_t+1) will induce cumulative model bias.

does not convince me. First of all, I do not see the cumulative model bias, because you are still using only a one-step prediction. Second, what is worse the model bias or the bad combination of state and action? An ablation study, looking deeper into this design decision would be great.

The experimental section focuses purely on benchmarking the algorithm compared to existing algorithms in terms of the learning curve. These curves show that the Mave learning curve is higher than the others. I would wish for more experiments that look deeper into what is going on in the algorithm. There would be many different questions, how big is the difference between the on- & off-policy targets? Is the off-policy target an underestimation of the critic or an overestimation? In my opinion, the learning curve is above your learning curve experiments do not contain much information anymore.

**Summary Of The Paper:**

Summary:
The paper proposes to mix off- & on-policy critic updates to improve sample efficiency and increase exploration. More specifically, the critic loss function is L = (k * TD_Q_off_policy + (1-k) * TD_Q_on_policy)**2. To enable the on-policy updates, a reward and dynamics model is learned from the collected data. In the experiments, the paper compares learning curves averaged over 10 seeds on the standard openAi gym continuous control tasks. The proposed algorithm performs better than the baselines in terms of sample efficiency as the reward goes up higher faster.



**Summary Of The Review:**

* Conclusion:
The idea of the paper is not bad but also not innovative. I am happy that somebody tried mixing the off- & on-policy critic targets and reported their results on it. My personal takeaway from the paper is, that the off-policy nature of the critic targets does not seem to be a major limiting factor in the general approaches. Using the mixed critic target, one only gains 10-20% of performance. Unfortunately, the paper does not provide much insight into how this improved performance is obtained or what the limiting factors are. Is it the model error or not? Why is the a_t+1 chosen w.r.t. to the sampled and not the predicted state? Therefore, the paper is most likely not going to influence the trajectory of the community which is fine and could still be presented at ICLR. From a technical perspective, the paper is a borderline accept paper. However, in combination with the bad presentation and the need for a major revision the paper falls below the acceptance threshold and is a borderline reject for me.

---

> ### Comment · Reviewer_Zou8 · 2022-11-18
> **Post-Rebuttal Response**
>
> So far the authors have not addressed the shortcomings in comments or with an update. Therefore, I am reducing my score to 3.

---

### Official Review · Reviewer_3k6t · 2022-10-24

**Confidence:** 2
**Correctness:** 2
**Technical Novelty And Significance:** 3
**Empirical Novelty And Significance:** 2
**Recommendation:** 3

**Clarity, Quality, Novelty And Reproducibility:**

As discussed above, the clarity of the paper requires some work.

The proposed technique seems novel, interesting and promising.

**Strength And Weaknesses:**

### Strength
The use of learned transition probability and reward in constructing a Q function target is novel and interesting. This is relevant to the RL community. Furthermore, the experimental results seem promising.

### Weakness

The theoretical analysis does not seem to provide sufficient evidence for answering the major claim of this paper – this method can provide effective target value exploration and improve sample efficiency. The analysis (Theorem 2) bounds the error in Q function estimation by the errors in the learn models, but they do not necessarily show that the proposed method is more sample efficient. It’s unclear to me whether the proposed method is better than random exploration. Off-policy methods such as DDPG or SAC work just fine if the data collected have good coverage.

The clarity of this paper can be improved. The lack of clarity leads to large friction in understanding the correctness and contributions.

- I don’t quite follow some claims throughout the text. For example
    - “By instinct, the model-based target value is able to explore the future states from different view-points with some certainty”
    - “Fourth, the accuracy of the learned model is tested by setting a maximum online time step, which is the beginning of off-line planning that is isolated from the environment.”
- The problem setting has not been very clear. Is this an off-policy setting? What data is given? Is interacting with the environment allowed?
- The math derivation is not rigorous and the notation is sometimes confusing. Some examples:
    - In Eq (5), randomness in s_{t+1} and s_{t+1}^p are not considered.
    - Above Section 4, J(\theta), but the right hand size does not have \theta.
    - P^t under Eq (1) should be a function of \pi.
    - In Eq (3), what’s the distribution of s_{t-1}?
- The number of Lemma and equations are mislabelled throughout the text.

Other issues:
- The layout of the experimental section can be improved. It seems to be too much space taken by 5 figures in Page 8.
- In Lemma 1 and Lemma 3, should the expected KL-divergence be total variation distance?
- In Lemma 3, Eq (7) neglects the RL discount factor in the rates, which is important to understand how the rate is dependent on horizon.
- Using “dynamics” include both transition probability and reward function is a bit confusing.
- "planning" usually involves relatively long horizon optimization, I did not connect one-step prediction to planning.


**Summary Of The Paper:**

This paper proposes a model-based value exploration technique to improve actor-critic RL algorithms. This approach involves training models for transition probability and rewards. By one-step prediction using the models, this approach provides effective target value (Q function) exploration. By theoretical analysis and empirical results, the authors show that this technique is effective.

**Summary Of The Review:**

Due to lack in clarity and significance in theoretical analysis, I recommend rejection. But the direction seems interesting.

---

> ### Author Response · Authors · 2022-11-05
> **Response**
>
> This paper is half theoretical and half heuristic. So I can only proves some aspects. Sample efficiency is extremely difficult to be justified. As I   see, most related papers can only observe the sample efficiency from experiments. If there is, please show me some closely related references. Why my experiments do not have good coverage? I have employed five tasks and done abandon studies.
>
> The problem setting is that interacting with the environment allowed. I made the claim “Fourth, the accuracy of the learned model is tested by setting a maximum online time step, which is the beginning of off-line planning that is isolated from the environment.” to do some tests after the training is over.
>
> The claim of "randomness in s_{t+1} and s_{t+1}^p are not considered in Eq (5)" may be because you did not notice the extended explanation following Eq (5), which is starting with "where".  P^t under Eq (1) is also explained following Eq (1). Eq (3) is just a background information, whose distribution needs to be talked about in specific application of this information. J(\theta) is my mistake.

---

### Official Review · Reviewer_WVWG · 2022-10-25

**Confidence:** 4
**Correctness:** 3
**Technical Novelty And Significance:** 2
**Empirical Novelty And Significance:** 2
**Recommendation:** 3

**Clarity, Quality, Novelty And Reproducibility:**

The quality and clarity of the paper could be improved. The algorithm is novel and should be reproducible with the given information.

**Strength And Weaknesses:**

Better value estimated in actor-critic setting is an important area of research. Using model-based approaches to get better TD-targets is an interesting direction to improve such algorithms. However, the paper has several weaknesses. The writing should be improved. There are many grammatical incorrect sentences making it a little difficult to follow the paper. Although the proposed method itself is simple it requires much more effort than necessary to understand what the authors propose. For example, first stating the equation for the loss and then in another equation the resulting gradient (eq. 9 and eq.11) seems unnecessary to me and only takes space. Figure 1 showing the mujoco environments is also not needed and the plots in Figure 2 could be arranged much more space efficient. This would allow the ablation studies to be in the main paper instead of the appendix.

Conceptually, I do not understand were the advantage of MAVE should come from. As the dynamics model is trained on the same distribution of data that is also used to train the value function, the estimated next state from the dynamics model should just be a noisy version of the true next state which is used for the classical TD-target. An appropriate baseline would then be to use MAVE but to replace the next state estimated by the dynamics model with the true next state plus some additional noise. This would be similar to how TD3 adds noise to the action at the next state but instead adding noise to the next state.

The experiments show slight improvements over the baselines, however the reported results for some baselines are worse than in the literature - especially for HalfCheetah. This is also true for the v3 versions in gym (cf. https://spinningup.openai.com/en/latest/spinningup/bench.html). The authors likely mean the gym v3 versions when they speak about mujoco version 3, as there is no mujoco version 3.


**Summary Of The Paper:**

The paper studies off-policy actor-critic reinforcement learning. It proposes the MAVE algorithm where an additional dynamics model is learned. In the TD-target of the value function is defined as a mixture of the usual TD-target computed from the next state observed during collection of the experience and the TD-target computed from the next state estimated by the learned dynamics model from the previous state. In the experiments slight improvement over baselines are demonstrated.

**Summary Of The Review:**

Overall, the paper goes into an interesting direction but needs a lot of improvements to meet the acceptance threshold.

---

> ### Author Response · Authors · 2022-11-05
> **Response**
>
> Everyone has his preference on the layout, so I should put aside the problem of layout arrangement and talk about the contents.
>
> Indeed, the dynamics model is trained on the same distribution of data, but the dynamics model is a distribution itself, which changes the distribution of next state. The estimated next state from the dynamics model is noisy, which can be seen as exploration. In other words, the estimated next state provides more possible ways from the view of heuristic thinking.
>
> Noisy exploration is totally different from the true next state plus some additional noise, because the source of noise is different. In Mave, the noise comes from the inaccurate trained model. So I really do not understand why you equal my work with TD3. The noise from the inaccurate trained model will be minimized at the end of training.
>
> The augument of slight improvements over the baselines seems unfair because I can see dictinct improvements over my baselines. As for the comparison with the results of the literature, indeed I find it difficult to reproduce the results of other papers. But I point out in my experiment part that there are also some excellent papers who cannot reproduce the results of gym, either. I do not understand why it is so harsh for me.
>
> I have responsed all concerns.

---

> > ### Comment · Reviewer_WVWG · 2022-11-17
> > **Response to the Authors**
> >
> > After reading the other reviews and the authors reply I stick with my original evaluation of the paper.

---

### Decision · Program_Chairs · 2023-01-20

**Decision:**

Reject

**Justification For Why Not Higher Score:**

All reviewers believe that the paper should be rejected.

**Justification For Why Not Lower Score:**

N/A

**Metareview: Summary, Strengths And Weaknesses:**

Although some of the reviewers agree that there are novelties in this work, none of them believe that the paper is ready to be accepted. Among issues that were raised are:
- The writing quality should be improved to make the paper more understandable.
- The theoretical justifications do not provide sufficient evidence for the claims of the paper.
- The empirical results do not carefully study the reasons for performance improvement.
- Connections to prior work should be improved.

Please incorporate the reviews to improve your work.